# Multiple signal classification algorithm for super-resolution fluorescence microscopy

Krishna Agarwal[1] & Radek Macháň[2]

Single-molecule localization techniques are restricted by long acquisition and computational times, or the need of special fluorophores or biologically toxic photochemical environments. Here we propose a statistical super-resolution technique of wide-field fluorescence microscopy we call the multiple signal classification algorithm which has several advantages. It provides resolution down to at least 50 nm, requires fewer frames and lower excitation power and works even at high fluorophore concentrations. Further, it works with any fluorophore that exhibits blinking on the timescale of the recording. The multiple signal classification algorithm shows comparable or better performance in comparison with single-molecule localization techniques and four contemporary statistical super-resolution methods for experiments of *in vitro* actin filaments and other independently acquired experimental data sets. We also demonstrate super-resolution at timescales of 245 ms (using 49 frames acquired at 200 frames per second) in samples of live-cell microtubules and live-cell actin filaments imaged without imaging buffers.

[1] BioSystems and Micromechanics Inter-Disciplinary Research Group, Singapore-MIT Alliance for Research and Technology, 1 CREATE Way, 04-13/14 Enterprise Wing, Singapore 138602, Singapore. [2] Department of Biological Sciences and Centre for BioImaging Sciences, National University of Singapore, 14 Science Drive 4, Singapore 117543, Singapore. Correspondence and requests for materials should be addressed to K.A. (email: uthkrishth@gmail.com).

Super-resolution fluorescence microscopy techniques aim at resolving details smaller than the Abbe diffraction limit of $\frac{\lambda}{2\text{NA}}$, where $\lambda$ is the wavelength of the fluorescence emission and NA is the numerical aperture of the microscope objective. Most of these techniques use the blinking phenomenon, where fluorophores switch between a bright (fluorescent) state and a long-lived dark state. A series of images is recorded over time. Each image has different intensity distribution because a different set of fluorophores were in the bright state during each image acquisition. The temporal information contained in the series is, then, used to construct a final image with improved spatial resolution. Single-molecule localization microscopy (SMLM) techniques such as stochastic optical reconstruction microscopy (STORM) or photo-activated localization microscopy (PALM) are popular super-resolution techniques owing to their simplicity, few (if any) special requirements on instrumentation, and impressive resolution of ∼20 nm (refs 1–3). However, they require that the fluorophores exhibit long dark states, so that only a small subset of optically separable fluorophores are in the bright state in each frame of the image stack. This translates into requirements of long acquisition times and of photochemical environment promoting long dark states and impeding bleaching, which is toxic to live cells[4].

The limitations of SMLM have motivated development of techniques that rely on statistical independence of blinking of individual fluorophores rather than on long dark states[5]. Such techniques include super-resolution optical fluctuations imaging (SOFI[6]), Bayesian analysis of blinking and bleaching (3B (ref. 7)) and entropy-based super-resolution imaging (ESI[8]). Although they relax the requirements of SMLM, they do not reach resolution achievable by SMLM (∼110 nm for SOFI[9], 80 nm for ESI[8] and 50 nm for 3B (ref. 7)) and they possess limitations of their own. For example, SOFI uses cumulants of the fluorescence blinking to enhance resolution; since cumulants of orders higher than six are prone to shot noise and do not have good approximations, the practically achievable resolution improvement is limited to the factor of $\sqrt{6}$ (ref. 6). 3B uses a Markov process for modelling the blinking and bleaching of the fluorophores and an expectation maximization approach to determine the likelihood of an emitter being present at a given location. This approach is computationally intensive and its convergence to global minimum is not guaranteed.

In the following, we propose a novel algorithm utilizing fluorescence blinking to enhance spatial resolution. The algorithm, called MUltiple SIgnal Classification ALgorithm (MUSICAL), achieves super-resolution by exploiting the eigen-images of the image stack, which statistically represent its prominent structures and, then, applying the knowledge of the point spread function (PSF) of the imaging system to localize the structures to super-resolution scales. Like SOFI or 3B and other related techniques, MUSICAL requires neither special instrumentation nor special fluorophores. The sole requirement is statistically independent blinking of individual emitters. We tested MUSICAL on images of actin filaments and compared it with STORM, showing that both techniques give comparable resolution enhancements. We also compared MUSICAL with 3B, SOFI, ESI and deconSTORM[10] on experimental data sets independently acquired by other super-resolution research groups[11,12] and show comparable or superior performance of MUSICAL. We also demonstrate that MUSICAL performs well in situations where STORM fails due to high density of fluorophores. Further, we show that MUSICAL can be used for live-cell fast imaging (∼49 frames amounting to a total acquisition time of less than 250 ms) of live cells expressing standard green fluorescent protein (GFP) imaged in physiologically conducive buffer devoid of chemicals that influence blinking.

## Results

**Multiple signal classification algorithm.** The idea of MUSICAL is inspired from MUltiple SIgnal Classification (MUSIC) used in acoustics[13], radar signal processing[14] and electromagnetic imaging[15] for finding the contrast sources created due to scattering and contributing to the measured signal. However, MUSICAL differs from MUSIC because the emitters in fluorescence microscopy behave differently from the contrast sources encountered in scattering. Firstly, the fluorophores exhibit intermittent emission when exposed to continuous excitation, the intermittence patterns of any two fluorophores being uncorrelated. Secondly, the information of a molecule is concentrated in a small region defined by the PSF in the image plane. Thus, there is a region of confidence for the likely position of each molecule; pixels beyond that region contribute only additional noise and almost no usable information in the context of localizing molecules in the given region. Thirdly, MUSIC in its traditional form needs a higher number of receivers (pixels in the camera) and measurements (frames) than the number of contrast sources (fluorescent emitters). Thus, when imaging an area with a high density of fluorophores, the required number of frames may easily become impractically large when using the traditional form of MUSIC.

MUSICAL overcomes those issues of MUSIC by adopting a sliding soft window and stitching approach. MUSIC is applied only on a small part of the image (the soft window) at a time. The soft window is, then, scanned over the whole image and the individual MUSIC reconstructions are stitched together to form the MUSICAL image. See Supplementary Fig. 1 for the flowchart of MUSICAL. Since the area corresponding to the soft window contains only a fraction of the total number of fluorophores in the imaged area, the required number of frames in the stack is considerably reduced. Besides, this approach also suppresses the problem of noise contribution from pixels further away from a given emitter location. The concept, algorithm, and implementation of MUSICAL is detailed in Supplementary Notes 1 and 2, and Supplementary Methods, respectively. Several synthetic examples have been used to demonstrate various aspects of MUSICAL. Their details are given in Supplementary Methods. We discuss here the algorithm of MUSICAL briefly.

Using the selected soft window of the image stack, MUSIC first computes the eigenimages of the soft window through singular value decomposition. The effect of the soft window is illustrated in Supplementary Fig. 2. Each eigenimage of the selected soft window is associated to a singular value and represents a particular pattern found in the image stack. Statistically, large singular value indicates that the pattern of the corresponding eigenimage is a prominent pattern in the image stack. On the other hand, statistically less likely patterns, which include noise patterns, are characterized by small singular values. Thus, the eigenimages can be separated into two sets (or subspaces) using a threshold value $\sigma_0$, where eigenimages with singular values more than or equal to $\sigma_0$ form the signal subspace (or the range) and the remaining eigenimages form the null subspace. The concepts of the range and the null space are discussed in Supplementary Note 3. The choice of the value of $\sigma_0$ is discussed in Supplementary Figs 3 and 4, and Supplementary Note 4.

Then, the projections of the PSF (an example is given in Supplementary Fig. 5) at a given test point on the range and the null subspace are determined, which we denote as $d_{PR}(r'_{test})$ and $d_{PN}(r'_{test})$ respectively. The projection $d_{PR}(r'_{test})$ indicates whether the PSF at the test point is related to the patterns represented by the range, similarly for $d_{PN}(r'_{test})$. If the test point indeed belongs to either the fluorophore locations or the structure represented by them, then $d_{PN}(r'_{test})$ is close to zero, it is non-zero otherwise.

This property of $d_{PN}(r'_{test})$ is used to compute an indicator function as follows

$$f(r'_{test}) = \left(\frac{d_{PR}(r'_{test})}{d_{PN}(r'_{test})}\right)^{\alpha}. \qquad (1)$$

such that the value of the indicator function is very large at the point a fluorophore is present and small at the point where no fluorophore is present. An example of eigenimages and projection of PSF on them is given in Supplementary Fig. 6 and Supplementary Note 5. Further, the sensitivity of MUSICAL to the estimate of PSF is given in Supplementary Fig. 7.

In the original form of MUSIC, as used in acoustic and electromagnetic imaging, only the distance from the null space $d_{PN}(r'_{test})$ is used and $\alpha$ is often set as 1. However, the indicator function of MUSIC is not suitable for the sliding window approach, as shown in Supplementary Fig. 8. The modified indicator function of MUSICAL is amenable to the sliding window approach since the inclusion of $d_{PR}(r'_{test})$ in the numerator automatically weighs the result of each window, so that stitching of the results for sliding windows can be simplified as sum of the indicator functions of all the sliding windows covering a test point. At the same time, the use of $\alpha$ allows for better resolution by non-linearly scaling the indicator function at closely located test points. See Supplementary Note 6 for a discussion on the role of the parameter $\alpha$ and Supplementary Fig. 3 for an example.

**In vitro experiments**. Here, we present results of MUSICAL for *in vitro* experiments and *in vitro* test data available online[11,12]. *In vitro* experiments were performed on actin filaments tagged with Phalloidin-Atto 565 dye and serve to characterize the performance of MUSICAL. Results of samples 1 and 2 are included in Figs 1 and 2, respectively.

Experimental data provided as single-molecule localization test data set (referred to as Data-SMLM)[11] are used for comparison

with other contemporary methods. Details about this data set can be found in its website. It comprises of three data, namely tubulins long sequence (15,000 frames), tubulins high density (500 frames) and tubulinAF647 (9,990 frames). The results for Data-SMLM are included in Fig. 3. Tubulins high-density data is further used to demonstrate resolution of two microtubules using MUSICAL.

We breakdown the *in vitro* results into the following studies: characterization of the PSF and resolution of MUSICAL, performance of MUSICAL for non-sparse blinking, minimum number of frames required for MUSICAL, excitation power and MUSICAL, comparison of MUSICAL with SMLM techniques and comparison of MUSICAL with other super-resolution techniques. These studies are presented below.

**Point spread function and resolution of MUSICAL**. *In vitro* sample 1 is used for characterization of the experimental PSF of MUSICAL. In Fig. 1c, group of line segments AA is used to obtain the experimental PSF of MUSICAL. The lines are separated by 26 nm. The maxima of the profiles of all the lines are aligned and then their average profile is computed as representative of the PSF of MUSICAL, which is shown in Fig. 1l. The full-width at half-maximum (FWHM) of the average profile is 47.4 nm. We also determined FWHMs of all the individual profiles; they were $47.3 \pm 7.4$ nm (mean ± s.d.). We repeated the same procedure for another group of line segments BB shown in Fig. 1g. The lines are separated by 65 nm here. The FWHM of the average profile for group BB is 46.6 nm. The FWHMs of the individual profiles were $48.5 \pm 8.7$ nm (mean ± s.d.). Characterization of the PSF of MUSICAL as a function of signal to background (SBR) ratio using a synthetic example SynEx3 is given in Supplementary Fig. 9a.

We observed periodicity along the actin filaments in the MUSICAL images and the minimum distance between the two peaks on actin filament to be 27.7 nm, as seen in Fig. 1d. Further,

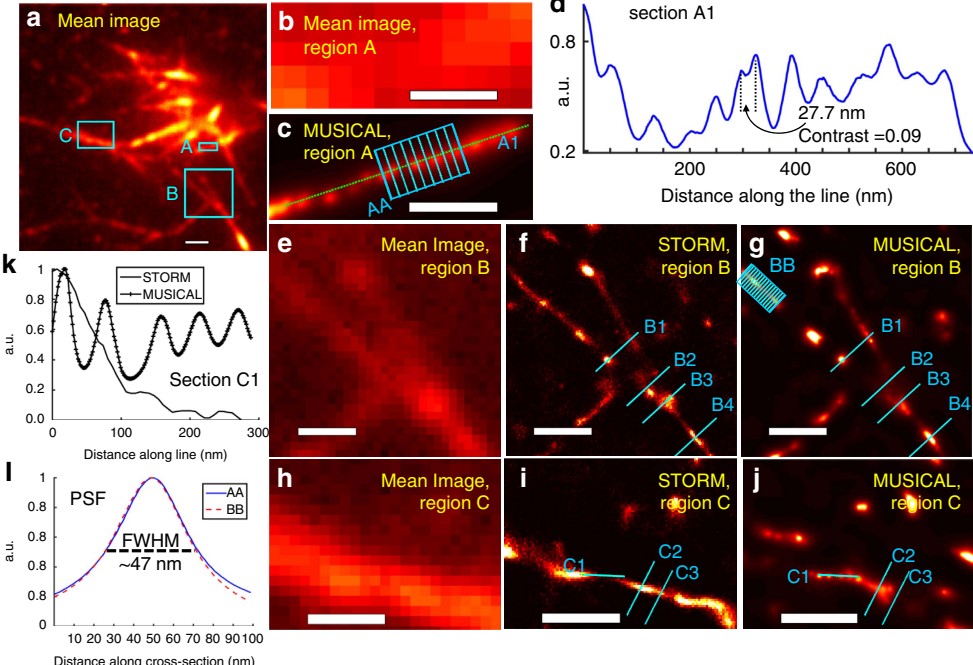

**Figure 1 | Results for regions A–C of *in vitro* sample 1.** The mean image of 10,000 frames is shown in **a**. (**b,c**) The mean image and MUSICAL image for region A. (**d**) Shows the profile of section A1 shown in **c**. (**e–g**) The mean image, STORM image, and MUSICAL image for region B. (**h–j**) show the mean image, STORM image and MUSICAL image for region C. (**k**) shows the profile of section C1 shown in **i,j**. (**l**) shows the average PSFs derived from groups AA and BB of line segments shown in **c,g**, respectively. Scale bars: 1 μm (**a**); 250 nm (**b,c,h–j**); and 500 nm (**e–g**).

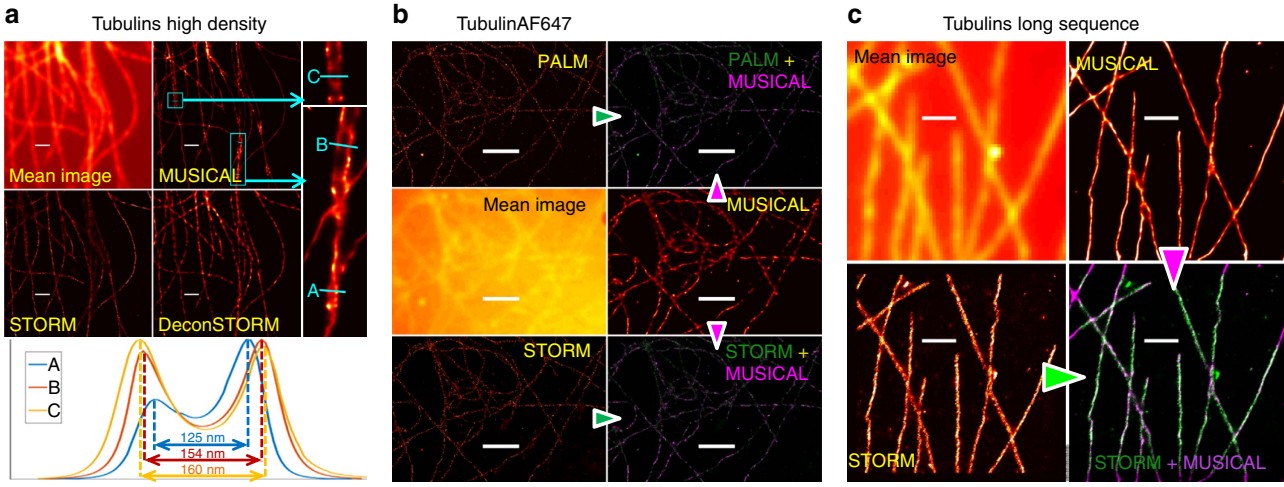

**Figure 2 | MUSICAL results for *in vitro* sample 2.** The first, second and third columns correspond to the image stacks acquired using laser powers 10.3 W cm$^{-2}$ (**a,e,i,m,q**), 40.2 W cm$^{-2}$ (**b,f,j,n,r**) and 205.6 W cm$^{-2}$ (**c,g,k,o,s**), respectively. The fourth column (**d,h,l,p,t**) shows result of MUSICAL using 49 frames only from the image stack acquired with 205.6 W cm$^{-2}$ excitation power. (**q–s**) Compares the intensity profiles of mean image and MUSICAL at the sections E1, E2 and E3, respectively. (**t**) Compares MUSICAL result at E3 using entire stack of 10,000 images with MUSICAL result at F3 using 49 frames only. Scale bars: 2 μm (first and second rows); and 1 μm (third and fourth rows).

**Figure 3 | Results on Data-SMLM data sets and synthetic example SynSTORM for comparison of MUSICAL with SMLM techniques.** (**a**) Tubulins high density (500 frames), (**b**) tubulins long sequence (15,000 frames) and (**c**) TubulinAF647 (9,990 frames). PALM and STORM results in **b** are taken from[11]. In **b**, MUSICAL image (in green) is overlaid with PALM image (in magenta) and STORM image (in magenta), respectively, for comparison. Similarly, MUSICAL image (in magenta) is overlaid with STORM image (in green) in **c**.

we consider the tubulins high-density data from Data-SMLM data set which has high density of emitters such that blinking is non-sparse. MUSICAL clearly illustrates the ability to resolve microtubules separated by 125 nm with a contrast value of 0.39, as seen for the section A in Fig. 3a. Characterization of structural resolution of MUSICAL is given in Supplementary Fig. 9b–d using synthetic examples SynPairDelX.

To study further the observed periodicity in the actin filaments, we plot the intensity of MUSICAL image across the sections A1 and C1 of two actin filaments in Fig. 1d,k, respectively. Fourier spectra of these sections and a synthetic example with periodically placed emitters are provided in Supplementary Fig. 10. It reveals a prominent peak at the spatial frequency corresponding to a period of 62.5 nm for A1. Another prominent peak at similar length scale occurs at sampling frequency corresponding to a period of 91.0 nm (which is $\sim$1.5 times 62.5 nm). Similarly, for the section C1, the Fourier analysis revealed a prominent peak at the spatial frequency corresponding to a period of 65.5 nm.

Chiu et al.[16] reported the period of the actin double helix to be 37 nm, which corresponds to a period of 74 nm for a single helix. However, actin is known to have structural polymorphs[17] and disorders of sub-helical dimensions[18]. In addition, poly-lysine, used for stabilizing the actin filaments on the glass chamber, is known to introduce polymeric variations[19]. Thus, we suggest that this periodicity is associated with the periodicity of a single helix of the double helical structure.

**The performance of MUSICAL for non-sparse blinking**. As noted before, most localization approaches rely on long dark states of fluorophores (sparse blinking) such that the few bright fluorophores are optically resolvable. MUSICAL does not have this restriction. To illustrate this, we consider *in vitro* sample 2 which exhibits high density of fluorophores (see Supplementary Fig. 11). While STORM is not suitable for imaging such sample (see Supplementary Fig. 12), MUSICAL can image these samples efficiently as seen in the second and fourth rows of Fig. 2).

We further demonstrate the ability of MUSICAL to image dense fluorophore structures through the tubulins high-density data of Data-SMLM. The results are shown in Fig. 3a. The zoom-in of the cyan marked rectangles in the MUSICAL results show that closely lying microtubules with separation smaller than the optical diffraction limit ($\sim$265 nm for this data) can be resolved consistently using MUSICAL.

**Minimum number of frames required for MUSICAL**. If blinking is sparse, large number of frames have to be acquired to be able to collect fluorescent signal from each fluorophore. However, since sample 2 does not demonstrate sparse blinking, it is interesting to consider if MUSICAL can be applied to such sample using only a few frames.

Owing to the sliding window of size $N$ pixels, the rank of the matrix on which eigenimages are computed is less than or equal to $\min(N,K)$, where $K$ is the number of frames; see Supplementary Note 1. The number of pixels in the sliding window $N$ is determined by the PSF of the system and thus fixed. Thus, it is desirable to use $K \geq N$.

We have used a sliding window of size $N_w = 7$ pixels, which implies that $N = N_W^2 = 49$. We applied MUSICAL on the first 49 frames of the data acquired for *in vitro* sample 2 under excitation power 205.6 W cm$^{-2}$. The results shown in the fourth column of Fig. 2 clearly illustrate the capability of MUSICAL to obtain super-resolution in non-sparse blinking using very few frames. This also indicates the utility of MUSICAL in the study of dynamic biological processes.

We note that emitters that do not emit significant number of photons in the frames used for MUSICAL will not be imaged by MUSICAL. As a further comment on the desirable condition $K \geq N$, we note that soft window of size larger than $N_w$ in Supplementary Fig. 2 does not add much value while increasing the minimum number of frames and the computation time. Thus, the sufficient number of frames is limited by the blinking rate and emitter density such that all emitters in the sample can blink in order to represent the entire structure and there are enough fluctuations in the intensity over the frames.

**Excitation power and MUSICAL**. Whereas the image stack for sample 1 was acquired under 205.6 W cm$^{-2}$, we acquired three image stacks for sample 2, each with different excitation powers, that is, 10.3, 40.2 and 205.6 W cm$^{-2}$. On one hand, the excitation power directly impacts the achievable resolution through the signal strength and SBR ratio. On the other hand, it indirectly influences the reconstruction by affecting the blinking rates of fluorophores, where high excitation powers generally result in longer dark states[20].

The MUSICAL results for these image stacks are shown in Fig. 2. It is seen that at each excitation power, MUSICAL can reconstruct more details than the mean image. The FWHM of the peaks indicated by arrows in Fig. 2r,s are 49 and 61 nm, respectively. The result implies that the FWHM does not deteriorates significantly when the laser power is decreased from 205.6 to 40.2 W cm$^{-2}$. It also helps to identify the minimum power at which sufficient resolution is achieved while minimizing photo-bleaching. We provide an additional example using *in vitro* sample 3 and an empirical plot of MUSICAL FWHM as a function of excitation power in Supplementary Fig. 13 and Supplementary Note 7.

We discuss the presence of side lobes in experimental results of MUSICAL and provide an analysis of the side lobes of MUSICAL as a function of SBR in Supplementary Fig. 14. Effect of camera noise that becomes prominent at very poor SBR is discussed in Supplementary Fig. 15 and Supplementary Note 8. Compensation for camera noise and side lobes is discussed in Supplementary Fig. 16 and Supplementary Note 9. Compensation of side lobes of Fig. 2m is shown in Supplementary Fig. 17.

**Comparison between MUSICAL and SMLM techniques**. Comparison of MUSICAL and STORM results for regions B and C of sample 1 are shown in Fig. 1f,i. In addition, we compare the results of MUSICAL with SMLM techniques for the Data-SMLM data set in Fig. 3a–c. Implementation and parameters used for SMLM techniques are given in Supplementary Methods. Overlay of MUSICAL images with STORM or PALM images are also provided for tubulins long sequence and tubulinAF647. We chose these data for overlay and comparison because they demonstrate sparse blinking, and thus are favourable for SMLM. The results show good agreement between MUSICAL and SMLM techniques.

Now we discuss the difference between MUSICAL and SMLM results, such as noted in Fig. 1f,g,i,j. These differences are attributed to multiple simultaneously bright fluorophores in the small volume surrounding the junctions. If a couple of closely placed emitters are blinking in a frame in a very small volume, STORM may conclude the presence of one very bright fluorophore at the centre of the small volume instead of the actual distribution of the closely placed emitters, unless a multiple-fluorophore fitting approach is incorporated[5]. The differences are discussed more explicitly in the context of cross-section lines B1-B4 and C2, C3 in Fig. 1f,g,i,j in Supplementary Fig. 18, synthetic example SynFork in Supplementary Fig. 19 emulating the fork structure in Fig. 1f,g, and synthetic example

SynSTORM in Supplementary Fig. 20. Discussion on these results is given in Supplementary Note 10.

**Comparison with other super-resolution techniques.** Here, we compare MUSICAL with other super-resolution techniques, namely, SOFI[6], deconSTORM[10], 3B (ref. 7) and ESI[8], the implementation details of which are given in Supplementary Methods. All these methods use statistical analysis of the fluorescence intensity in the image stack instead of single-molecule localization. DeconSTORM combines expectation maximization and deconvolution to form a super-resolved image which converges to the maximum likelihood estimate of emitter locations. It can be used for non-sparse blinking as well. We present comparison of these techniques for region A of *in vitro* sample 1, tubulins high-density data in Data-SMLM, synthetic example SynEx2 and synthetic example with Poisson noise SynEx2SBR3 in Fig. 4a–d, respectively. Further, comparison of MUSICAL with these techniques on the 3B test data[12] is given in Supplementary Fig. 21. Heavy computational requirements and convergence issue of 3B were found to be limiting in generating the comparison results. Thus, instead of using the

entire image stacks, we have used lesser number of frames and smaller region for generating the results for all the methods. We have used only 300 frames from sample 1 and only 200 frames from the tubulins high-density data of Data-SMLM. SynEx2 and SynEx2SBR3 have 49 frames each. Overlaid images of results of two methods in magenta and green colours are included in Fig. 4a for the ease of comparison. Coloured triangles are used to indicate the respective colours used for the methods in the overlaid images.

For region A of sample 1 in Fig. 4a, MUSICAL and 3B perform similar and provide better result than deconSTORM, ESI and SOFI. For Data-SMLM, MUSICAL results in sharper images than 3B and deconSTROM, as seen in Fig. 4b. In Fig. 4b, the cyan arrows indicate an instance where MUSICAL is able to resolve two microtubules that are unresolved by 3B and deconSTORM. Lastly, we discuss the comparison of MUSICAL with other methods for synthetic examples SynEx2 and SynEx2SBR3 shown in Fig. 4c,d, respectively. The details of these synthetic examples are presented in Supplementary Methods. It is seen that MUSICAL gives the sharpest result for the noise-free image stack, followed by deconSTORM and 3B. However, deconSTORM

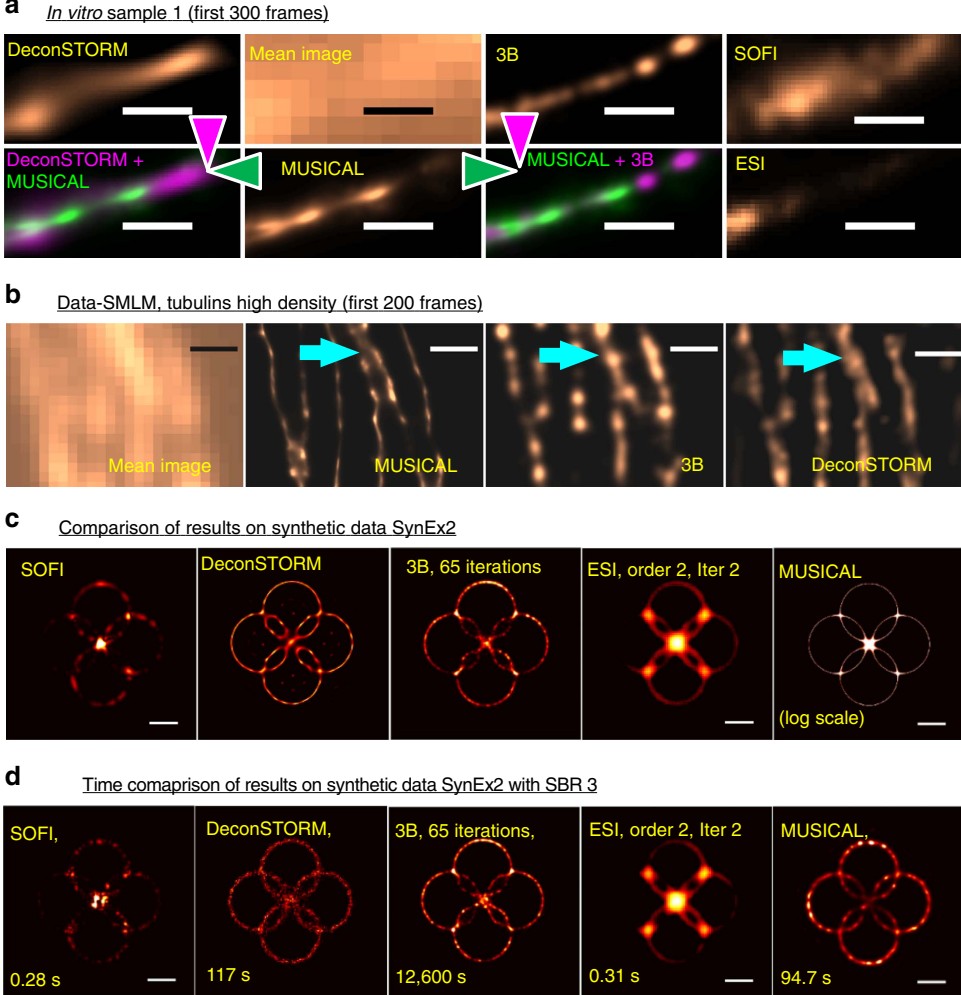

**Figure 4 | Comparison of MUSICAL with other super-resolution methods that perform statistical analysis of blinking statistics rather than SMLM.** (**a**) shows comparison for region A of *in vitro* sample 1. Overlay of MUSICAL image with 3B and deconSTORM is given for comparison in addition to their individual results. (**b**) Comparison of MUSICAL results with 3B and deconSTORM for the tubulins high-density data of Data-SMLM. MUSICAL can resolve two closely lying microtubules (cyan coloured arrow) which are unresolvable by 3B and deconSTORM. (**c**) Comparison results for synthetic example SynEx2 which is noise-free. (**d**) Shows comparison results for synthetic example SynEx2 with Poisson noise statistics and SBR ratio 3. Includes the computation time for each method below its result.

generates some artifacts inside the rings, which seem to be related to side lobes.

For SynEx2SBR3, deconSTORM, 3B and MUSICAL provide reasonable reconstructions, although 3B reconstructs finer rings than MUSICAL. However, 3B generates point-like artefacts along 45° and 135° directions. This might be the case because of the very low number of frames. DeconSTORM also generates artefacts in the regions where the rings are close to each other. ESI provides good reconstruction, although visible in the logarithmic scale only. We present a summary and comparison of features of various super-resolution techniques in Supplementary Table 1 and Supplementary Note 11. SynEx2SBR3 is used for comparison of the computation time also. The time taken by each method is reported below its result in Fig. 4d. While MUSICAL takes significantly larger computation time than SOFI and ESI, it compares well with deconSTORM and is orders of magnitudes faster than 3B. We note that no parallelization has been used for reporting the computation time of MUSICAL, although there is a scope for parallelization and thus improvement in the computation time of MUSICAL.

**Live-cell experiments.** Two sets of live-cell experiments were conducted on Chinese hamster ovary cells (CHO-K). The cells were imaged using total internal reflection fluorescence (TIRF) microscope under physiologically relevant conditions. In the first set of experiments performed on live-cell microtubules sample 1 and 2 (details in methods), we imaged microtubules labelled by transiently expressed GFP-tubulin. MUSICAL results for sample 1 are presented here.

In the experiment with live-cell microtubules, we used regular GFP and standard live-cell imaging medium with no additive chemicals to influence blinking. Thus, the cells were in physiologically relevant conditions optimally suited for biological experiments. A 488-nm laser excitation at the intensity of $25\,W\,cm^{-2}$ was used. In live cells, microtubules are highly dynamic structures and there is a continuous exchange of tubulin molecules between the microtubules and cytosol[21]. Therefore, there is a significant population of labelled tubulin molecules freely diffusing in the cytosol in addition to the labelled microtubules. To suppress the large background introduced by these freely diffusing molecules, we have heuristically used a lower bound of 0.4 times the MUSICAL intensity that corresponds to 99.9% of the histogram of intensities. This choice is analogous to the heuristic choice of the upper and lower bounds used in 3B (ref. 7). Further, only a part of the population of tubulin molecules in the cell is labelled. Thus, the microtubules are not uniformly labelled and fluorescent in parts only.

The MUSICAL result for live-cell microtubules sample 1 in Fig. 5 is generated using the first 49 frames, each with 5 ms exposure, which amounts to a total time 245 ms for a MUSICAL image. The zoom-ins shown in Fig. 5c,e indicate that MUSICAL provides resolution of less than 100 nm in the live cells too. We note that the overlay of the microtubules as seen in the zoom-in has been observed before[22,23] and not considered particularly significant. We use this structure to illustrate sub-100 nm resolution and the reconstruction of details by MUSICAL. This is further validated quantitatively in Fig. 5g, which plots the distance of the closest maximum along the direction $n_1$ (or $n_2$) from the cyan coloured line shown in Fig. 5d. It shows that the three microtubules can be identified separately even though the distance between them is quite small.

The effect and benefit of using lower number of frames in live-cell experiment is demonstrated in Supplementary Fig. 22, which shows results for region B of live-cell microtubules sample 1 with different frame numbers taken from different parts of the

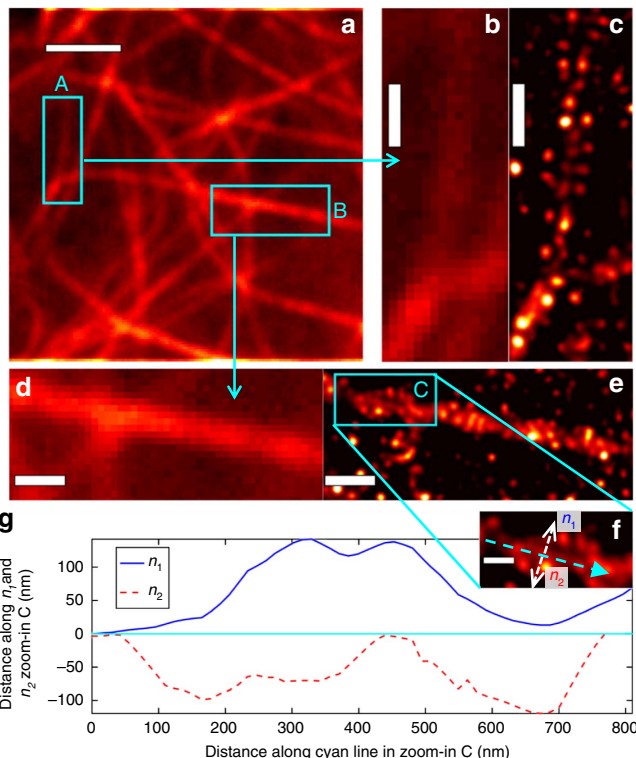

**Figure 5 | Result of MUSICAL for live-cell microtubules sample 1.** Mean image of 49 frames is shown in **a**. Mean image and MUSICAL result for region A identified in **a** are shown in **b**,**c**. Mean image and MUSICAL result for region B identified in **a** are shown in **d**,**e**. Zoom-in of region C identified in **e** is shown in **f**. Scale bar, 200 nm. Sub-figure **g** shows the closest maxima from points along the cyan line in **f** in the directions $n_1$ (blue) and $n_2$ (red).

whole-image stack. Further, we show using the results of live-cell microtubules sample 2 in Supplementary Fig. 23 that the timescales of MUSICAL may potentially be reduced to less than 50 ms.

In the second set of live-cell experiments, we imaged GFP-labelled actin in cortical cytoskeleton of a CHO-K1 cell stably expressing Lifeact-GFP. The results are given in Supplementary Fig. 24. It shows the ability of MUSICAL to image dense nanoscale structures.

**Discussion**

Although the reported resolution of STORM at 20 nm is significantly better than MUSICAL, MUSICAL conclusively demonstrates less stringent requirements on sparsity of blinking, density of emitters, number of frames, excitation power and chemical environment. Further comparison of MUSICAL with other statistical super-resolution techniques, such as SOFI, deconSTORM, 3B and ESI indicates competitive and often superior performance of MUSICAL. Here, we mention that MUSICAL performs better than SOFI and ESI in terms of resolution and image quality, however at the cost of increased computational time and using the knowledge of the PSF. The analysis of sensitivity of MUSICAL to the accurate knowledge of the optical PSF in Supplementary Fig. 7 indicates that MUSICAL is not too sensitive to PSF. Although 3B shows better robustness to noise than MUSICAL, it has significantly higher computational demands unless faster implementation of 3B using cloud computation[24] is used.

Lastly, to the best of our knowledge, none of the currently known super-resolution algorithms so far has demonstrated the

capability to produce super-resolved images of live cells in their natural environment with total acquisition times of the order of 50–250 milliseconds using regular non-sparse blinking dyes. We do note that Huang *et al.*[25] have reported super-resolution at video acquisition rate of 32 frames per second, although using photoswitchable proteins or Alexa Fluor 647 in glucose oxidase and catalase-based imaging medium, as opposed to the live-cell results reported here obtained using regular GFP and without oxygen scavenging and carbon dioxide-deficient imaging medium considered toxic to live-cells[4]. We believe that MUSICAL is a step forward towards the goal of fast super-resolved imaging of biological phenomena in their natural state.

## Methods

***In vitro* actin sample preparation.** We used the protocol suggested in ref. 26 for forming the *in vitro* actin samples. Preformed actin filaments from Cytoskeleton Inc., extracted from rabbit skeletal muscle, were used for forming the sample. In one chamber of an 8-chambered cover glass of ∼750 μl volume, 200 μl of 0.01% poly-L-lysine solution was incubated for 10 min. Remaining liquid was drained out using a pipette. A total of 90 μl of general actin buffer (reconstituted as suggested by Cytoskeleton's product datasheet of general actin buffer) was introduced in the chamber. This was followed by adding 10 μl of 10 μM preformed actin filaments and 10 μl of Phalloidin Atto-565 solution (stock solution prepared as recommended by the manufacturer, Sigma-Aldrich). The contents of chamber were mixed by gently pipetting up and down. An incubation time of ∼45 min was allowed. Then the liquid in the chamber was removed by pipetting. For sample 1, the contents of the chamber were washed 3 times using Buffer B (see details of imaging buffer below) by a two pipette system, where 1 pipette let in the buffer and the other pipette sucked out the solution. Sample 2 was washed five times. Sample 3, with results in Supplementary Fig. 13, was not washed at all. In sample 2, we introduced tetraspeck beads in the imaging solution and allowed the solution to settle for 3 hours. In samples 2 and 3, where the same samples are imaged with different excitation powers, the image stacks have been appropriately shifted to compensate for the drift of sample between different acquisitions. See Supplementary Table 2 for an overview of the *in vitro* samples.

**Preparation and introduction of imaging buffer.** We used the imaging buffer composition suggested in ref. 27. Buffer A composed of 10 mM of TRIS (pH 8.0) and 50 mM NaCl. Buffer B composed of 50 mM TRIS (pH 8.0), 10 mM NaCl and 10% Glucose (weight per volume). GLOX solution (1 ml) was formed by vortex mixing a solution of 56 mg of Glucose oxidase, 200 μl of Catalase (17 mg ml$^{-1}$) and 800 μl of Buffer A. MEA solution (1 M, 1 ml) was prepared using 77 mg of MEA and 1 ml of 0.25 N HCl. For one chamber of an eight-chambered cover glass of about 750 μl volume, 700 μl of imaging buffer was prepared by mixing 7 μl of GLOX solution, 70 μl of MEA solution and 620 μl of Buffer B on ice. This imaging buffer was used as the medium for *in vitro* actin samples. The chamber was filled completely and covered immediately to avoid replenishing of oxygen in solution.

**Live-cell sample preparation.** CHO-K1 cells were obtained from ATCC (Manassas, VA). Lifeact cells (CHO-K1 cells stably expressing Lifeact-GFP) were provided by Prof. Rachel S. Kraut (NTU, Singapore). CHO-K1 and Lifeact cells were cultivated in DMEM medium (Dulbecco's Modified Eagle Medium, Invitrogen; Singapore) supplemented with 1% penicillin G and streptomycin (PS, PAA, Austria), and 10% fetal bovine serum (Invitrogen; Singapore) at 37 °C in 5% (v/v) $CO_2$ environment. GFP-tubulin plasmid was a gift from Dr Pakorn T. Kanchanawong (MBI, NUS, Singapore). Electroporation was used for transfection of the cells, during which, 90% confluent cells in a 75 cm$^2$ flask were washed twice with 1 × PBS, trypsinized with 0.25% trypsin-0.03% EDTA solution for ∼1 min at 37 °C, and then re-suspended in culture medium. Cells were precipitated by centrifugation and re-suspended in small amount of resuspension R buffer (NeonTM Transfection System, Life Technologies, Singapore) and transferred half into one electroporation cuvette (2 mm wide, Bio-Rad; Hercules, CA) for one transfection. Between 300 and 500 ng ml$^{-1}$ of the plasmid were added. After electroporation pulse, cells were seeded back to prewashed cover glass (30 mm in diameter; Lakeside, Monee, IL) in a 35 mm culture dish. Transfected cells grew in the culture medium for 24–36 h before measurement. All cells were imaged in Phenol red free DMEM medium containing 10% fetal bovine serum. See Supplementary Table 2 for an overview of the live-cell samples.

**Imaging system and image acquisition.** The setup consisted of an inverted epifluorescence microscope (IX83, Olympus, Japan) equipped with a motorized TIRF illumination combiner (IX3-MITICO, Olympus, Japan) and a scientific complementary metal oxide (sCMOS) camera with 6.5 μm pixels (Orca-Flash4.0, Hamamtsu Photonics, Japan). A 488 nm laser (LAS/488/100/D) or 561 nm laser (LAS/561/100, Olympus, Germany) was connected to the TIRF illumination combiner in which the incidence angle was adjusted to give 110 nm penetration depth of the evanescent field. The 488 and 561 nm lasers were used for imaging live-cell samples or *in vitro* actin filaments, respectively. A × 100, numerical aperture 1.49 oil immersion objective (UAPON, Olympus, Japan) was used to illuminate the sample and collect the fluorescence image. The fluorescence light then passed through a major dichroic (ZT405/488/561/647rpc, Chroma Technology, Bellows Falls, VT) and a band-pass filter (ZET405/488/561/647m, Chroma Technology, Bellows Falls, VT).

The intensity of the excitation light was determined as the power of the laser light exiting the objective divided by the illuminated area; the power of the laser light was regulated by the proprietary laser control software and by an additional OD 1 neutral density filter. The camera was controlled by Micro-Manager 1.4 (ref. 28). For multiple power measurements, the first measurement is done using the lowest power and the power is subsequently increased. The optical PSF computed as an airy disk has FWHM of ∼198 nm for these parameters and emission wavelength 593 nm of the phalloidin Atto-565 dye and about 176 nm for emission wavelength 512 nm of the lifeact-GFP.

For *in vitro* samples 1 and 2, 10,000 frames were acquired with exposure time of 5 ms (200 frames per second) For *in vitro* sample 3, 20,000 frames were acquired with exposure time of 10 ms (100 frames per second). For live-cell microtubule sample 1, 1,000 frames were acquired with exposure time of 5 ms (200 frames per second). For live-cell microtubule sample 2, 49 frames were acquired with exposure time of 1 ms (1,000 frames per second). For live-cell F-actin sample, 100 frames were acquired with exposure time of 1 ms (1,000 frames per second).

**Data availability.** The data that support the findings of this study and the source codes of MUSICAL are available at https://sites.google.com/site/uthkrishth/musical.

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

# ARTICLE

20. Dempsey, G. T., Vaughan, J. C., Chen, K. H., Bates, M. & Zhuang, X. Evaluation of fluorophores for optimal performance in localization-based super-resolution imaging. *Nat. Methods* **8,** 1027–1036 (2011).
21. Nogales, E. Structural insights into microtubule function. *Annu. Rev. Biophys. Biomol. Struct.* **30,** 397–420 (2001).
22. Bloom, W. & Fawcett, D. *A Textbook of Histology* (Chapman & Hallp, 1993).
23. AMOS, L. A. & Klug, A. Arrangement of subunits in flagellar microtubules. *J. Cell Sci.* **14,** 523–549 (1974).
24. Hu, Y. S., Nan, X., Sengupta, P., Lippincott-Schwartz, J. & Cang, H. Accelerating 3B single-molecule super-resolution microscopy with cloud computing. *Nat. Methods* **10,** 96–97 (2013).
25. Huang, F. *et al.* Video-rate nanoscopy using sCMOS camera-specific single-molecule localization algorithms. *Nat. Methods* **10,** 653–658 (2013).
26. Metcalf, D. J., Edwards, R., Kumarswami, N. & Knight, A. E. Test samples for optimizing storm super-resolution microscopy. *J. Vis. Exp.* **79,** e50579 (2013).
27. *STORM Protocol-Sample Preparation. Technical Report* (Nikon Corporation, 2013).
28. Edelstein, A. D. *et al.* Advanced methods of microscope control using µManager software. *J. Biol. Methods* **1,** e10 (2014).

## Acknowledgements

This research was supported by the National Research Foundation of Singapore through the Singapore-MIT Alliance for Research and Technology BioSystems and the Micromechanics Inter-Disciplinary Research Programme. We acknowledge the help of Ms Shuangru Huang with cell culture and transfection, the gift of GFP-tubulin plasmid from Dr Pakorn T. Kanchanawong, and the gift of Lifeact cells from Prof. Rachel S. Kraut. We thank the providers of the online data and source codes used in this paper. We acknowledge discussion with Dr Lu Gan, Dr Kalpesh Mehta, Dr Dilip K. Prasad, Dr Sreelatha Sarangpani and Miss Shi Hua Teo. We especially thank Dr Dilip K. Prasad for several technical discussions.

## Author contributions

K.A. developed the algorithm, wrote the software, and performed simulations and data analysis; K.A. and R.M. performed the *in vitro* experiments; R.M. performed the live-cell experiments.

## Additional information

**Competing financial interests:** The authors declare no competing financial interests.

**Publisher's note**: 

