## [Peer Review File · Nature Communications]

Reviewers' Comments:

Reviewer #1 (Remarks to the Author):

The authors develop a new SR reconstruction algorithm, MUSICAL and test it on simulated and experimental data. The algorithm is one of the class of algorithms which combine information from multiple frames like SOFI and 3B. The claimed performance of the algorithm is impressive - essentially it is claimed to have near 3B performance with dramatically improved speed. 3B is pretty near the gold standard for density/ resolution/ imaging speed in localization microscopy, but is algorithmically so slow as to not be generally usable. If the claims are valid, an algorithm which combines decent algorithmic speed with 3B performance is certainly exciting.

Essentially, as most the manuscript stands, many of the comparisons are not convincing - noted below, and certainly from reading the main text I was inclined to be very sceptical. However, hidden away in figure S19 is the most exciting result of the paper - a direct comparison of MUSICAL to the other multiple frame methods. In figure S19 MUSICAL does appear to be pretty much on par with 3B in terms of final image quality.

However I believe the concerns identified here are too serious to accept the manuscript in current form. Honestly, it seems like the authors have come up with a promising SR algorithm but are new to the field, and therefore are not yet fully familiar with the standards which have developed in terms of necessary evidence to prove an SR algorithm's performance. Hopefully if they address the recommendations below they will be able to make a strong and valid comparison of their algorithm to the current state of the art.

Specific comments:

- Since the key performance comparison is to the multi-frame methods rather than the single frame methods, examples of SOFI & 3B on experimental data side by side with MUSICAL must be presented, and should be the primary comparison. Also, Fig S19 should be moved to the main text
- The authors use rainSTORM, a low density (single emitter fitting) algorithm as the main comparison to MUSICAL. This comparison is completely inappropriate as LD algorithms are not designed for the conditions tested here (high density of emitters). For comparison between MUSICAL and localization methods, they should exclusively use one of the many 2D high density localization algorithms developed for these conditions, eg DAOSTORM (the 3D DAOSTORM implementation is easier to use), CSSTORM, deconSTORM etc - see the 2013 localization microscopy challenge (Sage et al, Nat Methods 2015) for a reasonably full list.
- Experimental super-resolution data is all of poor quality, showing disturbingly punctate labelling which makes it very hard to assess algorithmic performance on real samples. Authors should consult eg Whelan & Bell Sci Rep 2015 on proper STORM labelling, possibly use a more optimal STORM buffer see eg Olivier et al PLoS One 2013, and acquire and analyse more optimal test data. An online resource with several experimental datasets from well labelled samples is also available: <http://bigwww.epfl.ch/smlm/datasets/index.html>. Possibly you could also ask Susan Cox for her raw data used in the original 3B paper.
- The run time of MUSICAL vs 3B is not clearly compared. Since the main advantage claimed on pg 6 is that MUSICAL is basically 3B performance but faster, this is really important. In Fig S19, there is a table of run times with 3B as 72 s vs MUSICAL 94 s. I think 3B was run on different hardware (cloud computing) which is why it is quite fast, but this is not clearly discussed. It is important to see a proper comparison of run times on the same hardware.
- There is no discussion of performance of the algorithms as a function of emitter density (bright emitters/ μm^2) which is the key parameter in HD algorithm performance (see any of the HD algorithm papers mentioned above).
- The Shaevitz lab published an algorithm SCORE (Deng et al, PLoS One 2014), which on the surface seems reasonably similar algorithmically - principle component/ eigenvector analysis of multiple

frames to extract super-resolved image. I am curious about how the two algorithms differ. SCORE achieves only quite low resolution (~ 100 nm) rather than the high resolution claimed here.

- Line plots are not an ideal metric for resolution (as discussed eg in supplement of Legant et al Nat Meth 2016). I realize FRC is not necessarily straightforwardly applicable for non-localization based methods. Do the authors have any other alternative measures of resolution? What about a radial plot of the reconstructed image fourier transform - essentially the system MTF?
- The claim pg 6 "Lastly, we note that to the best of our knowledge, none of the currently known super-resolution algorithms so far has demonstrated the capability to produce super-resolved images of live cells in their natural environment with total acquisition times of the order of 50-250 milliseconds using regular nonspare blinking dyes" is wrong - see Huang et al Nat Meth 2013.

Reviewer #2 (Remarks to the Author):

In this manuscript, Agarwal et al. described a new algorithm named "Multiple signal classification algorithm" for analysis of high density PALM/STORM data.

MUSICAL decomposes the raw data frames from a soft window into eigenimages to separate signal and noises and from there, solve for the underlying structure using both the signal subspace and the null subspace (noise). Final image is obtained by calculating a metric - the ratio between sub-sampled PSF projections into these two subspaces respectively.

The method resembles another version of noise reduction + deconvolution. In addition, I did not see how this method would provide single molecule location information which is the key for high resolution PALM/STORM images. I am not sure how the blinking behavior of the single molecules helped here except that it allows the authors to apply a soft window on image sequences to improve the computation speed. Another point is, taking the ratio between the projections in the two subspaces will artificially shrink the FWHM of structures and therefore, line profiles should not be used to demonstrate resolution achieved. Overall, I find this manuscript quite well written and interesting to experts in high density localization PALM/STORM algorithms. Considering the significant image artifact and lack of proper demonstration in its resolving power, I am not convinced whether it would be useful to PALM/STORM at its current state.

My other concerns are listed as below,

1. As mentioned above, line profiles should not be used to quantify resolution in highly processed images. For example, you can get a sharper profiles by taking a square of a confocal image. The "squared" images look sharper but it will not resolve features that are previously unresolvable. Therefore to demonstrate the resolution achieved, the authors should provide super-resolution images resolving features that are previously obscured by diffraction, for example, two sides of microtubules, mitochondria cristae, or vesicles.
2. The MUSICAL images from Figure 2a-2, there's a dark region around bright filaments. Is this real or is image artifact. Images from Figure 2 b2,c2,d2 of actin filaments are quite different from what others have shown using PALM/STORM (Nat Methods. 2012 Feb; 9(2): 185-188.). Is there a reason why it would look different from images that others have reported? Figure S18c also showed image artifact that were not part of the original signal. Would there be a way to decrease the large amount of artifact introduced?
3. Equation 3 in the supplement is derived based on the assumption that the pixel size of the imaging system is much smaller than the PSF width. This might not be true as many PALM/STORM systems including commercial systems use a pixel size that are close to the half width of the PSF.
4. The authors should double check the usage of the phrase "in vivo" in the context of the manuscript. Would "live cell" be more accurate?

Reviewer #3 (Remarks to the Author):

The manuscript under review introduces the traditional Multiple Signal Classification (MUSIC) in the signal processing community into the problem of single molecule localization microscopy (SMLM) and proposes a new algorithm, termed MUSICAL, to process SMLM data-set, featuring low requirements on laser power and number of frames, and successful reconstruction on denser fluorescent samples compared to STORM. By conducting a singular value decomposition on the measured intensity matrix, MUSICAL treats the set of eigen-images with small singular values as noise space and the set of eigen-images with larger singular values as signal space. Then, the authors project the Green function of each molecule onto the signal space and noise space, and evaluate the indicator function based on these projections to form a reconstructed image. To demonstrate the capability of this algorithm, the authors perform several in-vivo and in-vitro experiments with total acquisition time of 0.25 sec for a frame, which is very impressive for a SMLM technique.

Overall, the idea of applying MUSIC to SMLM problem is very interesting and novel. The authors carefully verify their results with reasonable simulations and the paper is well-written. However, there are some details of the algorithm is not clear to me and some questions as followed in this paper still need to be addressed before publication:

1. Some questions about MUSICAL:

- 1.1. It seems the authors pick the indicator function by intuition, is there other better way to do it?
- 1.2. In the description of α , the authors say that the larger the α the narrower the spread of the fluorescent molecule, which implies better resolution. However, the authors only choose $\alpha=4$ and stop increasing the number. Why not keep increasing to get even better resolution?
- 1.3. It seems to correctly estimate a threshold σ for signal noise separation is important to have a nice reconstruction. In the main paper, the authors estimate the SNR of sample 1,2 and in-vivo experiment. How do authors estimate SNR from the data? In addition, if we do not know the SNR, the authors say that they recognize the knee structure of the singular-value plot to set the threshold σ . However, Fig. S4 shows the reconstruction will have better resolution when you set the threshold even below the knee structure. I am wondering if you have a clear way to set the threshold for the case without SNR? Or the authors always estimate the SNR and set the threshold?
- 1.4. In the main paper, the authors use a sliding window of the size of 7×7 , which corresponds to the size of PSF, so the maximal rank of each patch is 49. This is why the authors only collect 49 images for reconstruction. I am curious about how the number of pixels for each window will affect the reconstruction. If the authors use a window size of 10×10 , for example, and take 100 images. Will the reconstruction be better than the previous case? It would also be interesting to see what would be the effect of number of frames to reconstruction in the simulation.
- 1.5. MUSICAL requires PSF to be known. Do the authors calibrate the PSF? What is the tolerance of the algorithm if the known PSF does not well match the experiment PSF?

2. Some questions of reconstruction comparison:

- 2.1. Following question 1.4, the last two columns of Fig. 2 in the main paper compared the reconstruction using 10000 frames and 49 frames. Although the reconstruction with 49 frames capture the main structure of the sample, the fine features seem to look different from the one with 10000 frames. Maybe some of the fine-structure information is not contained within these 49 images? It may also suggest MUSICAL does not have uniform resolution performance across the field of view?
- 2.2. For Fig. S7, it may seem that the reconstruction with 49 frames is sharper than the one with 200, 400, or 800 images. However, the one with 49 frames look not well-sampled (some parts of the sample is missing). Maybe 49 images are too few to have a nice reconstruction?

3. Minor details on the paper:

- 3.1. Supplementary section III C, the abbreviation SBR shows up before the definition in section III E.
- 3.2. Supplementary III D, is there any reason that authors choose different and for different

simulations?

3.3. Supplementary IV A, what does "0.4 times of the 99.9% of the histogram of MUSICAL intensities" mean? There are several descriptions like this.

Reviewer #4 (Remarks to the Author):

The authors propose a new algorithm for analyzing super-resolution images based on computing an eigenvector decomposition of a series of blinking images, followed by projection of the obtained images onto psf images. From the simulated data, the results look quite promising (with a few caveats - see below), although the application to real images is much less convincing.

I find the mathematical explanation of the method extremely hard to follow (with the caveat that I am not a theoretical mathematician, although I do believe that I'm more mathematically inclined than the typical nature communications reader). For this reason I believe that the basis of the method needs to be much more clearly explained if publication in this journal is to be considered. I particularly struggle with understanding what the range and null spaces are and what the eigenimages might look like. This could potentially be assisted with an additional supplemental figure showing examples of eigenimages which belong to each space.

What is particularly troubling for me is that I do not believe that the authors have clearly established WHY the performed manipulations are expected to give super-resolution. My very simplistic interpretation is that the eigenvector decomposition will, somewhat like independent component analysis (Lidke et al 2005), give a series of eigen-images which represent the individual molecules present, and that the projection onto the shifted copies of the PSF is somewhat like using an autocorrelation to find molecule peaks. The test function, however, seems fairly empirical, and the alpha parameter is not well justified. I find the alpha exponent very suspicious, as it seems to be chosen purely to make the images look sharper. You can make any fluorescence image look sharper by raising it to a power, but this would never be considered super-resolution.

In addition to a clearer explanation of the underlying mathematics, and reasoning as to why this should improve resolution, the authors should address the following points:

- One of the major issues with other non-localization based approaches (specifically SOFI) is that they are non-linear in emitter concentration. Is this method linear (some of the simulated data suggests that it might not be with the crossings of lines appearing more than twice as bright as a single line)?
- The width of a line profile (as used to quantify resolution on experimental data) is useless if the method is non-linear, and is generally somewhat suspect even for linear methods. Much better is to show that two lines can be resolved. The data in Fig1 on this does not show the two lines being clearly resolved at better than the diffraction limit.
- The resulting reconstructions of real data seem even spottier than PALM/STORM. Might this be a result of non-linearity, or is there another issue with the method?

Despite these criticisms, I believe that with a little more work on the explanation and a more extensive comparison to other methods this approach could address an important niche in the analysis of blinking based super-resolution - namely the analysis of dense data in a way that is a) better than SOFI and b) faster and less reliant on metaparameters than 3B.

Replies to the comments of Reviewer #1:

Dear Reviewer,

We sincerely thank you for a constructive and favorable review of our paper. Below, we reply to your comments and explain how and where we have addressed your comments in the paper.

Comment 1.a: However, hidden away in figure S19 is the most exciting result of the paper - a direct comparison of MUSICAL to the other multiple frame methods. In figure S19 MUSICAL does appear to be pretty much on par with 3B in terms of final image quality.

Comment 1.b: Since the key performance comparison is to the multi-frame methods rather than the single frame methods, examples of SOFI & 3B on experimental data side by side with MUSICAL must be presented, and should be the primary comparison. Also, Fig S19 should be moved to the main text

Reply: As recommended by you, we have moved Fig. S19 from the original supplement to the main text as Fig. 4(d,e). The computation times have been included in Fig. 4(e). Further, we have included comparison of MUSICAL with SOFI, 3B, and ESI using independent experimental data from the SMLM datasets and 3B test data, as suggested by you. We thank you for this valuable change in the manuscript.

*Please see the newly added **Fig. 4** on Page 8 and **magenta colored text on Page 7** of the revised paper.*

Comment 2: The authors use rainSTORM, a low density (single emitter fitting) algorithm as the main comparison to MUSICAL. This comparison is completely inappropriate as LD algorithms are not designed for the conditions tested here (high density of emitters). For comparison between MUSICAL and localization methods, they should exclusively use one of the many 2D high density localization algorithms developed for these conditions, eg DAOSTORM (the 3D DAOSTORM implementation is easier to use), CSSTORM, deconSTORM etc - see the 2013 localization microscopy challenge (Sage et al, Nat Methods 2015) for a reasonably full list.

Reply: We have included comparison with deconSTORM in the main paper in Figs. 3(a) and 4. We did generate results with CSSTORM as well but the results showed significant artifacts. We guess that we might not be using the right set of parameters or not providing the input data appropriate to the CSSTORM. Thus, we have not included the results of CSSTORM in the paper.

*Please see the newly added **Fig. 3(a) and Fig. 4** on Pages 6 and 8, respectively, of the revised main paper. Please see the **magenta colored text on Page 7** also.*

Comment 3: Experimental super-resolution data is all of poor quality, showing disturbingly punctate labelling which makes it very hard to assess algorithmic performance on real samples. Authors should consult eg Whelan & Bell Sci Rep 2015 on proper STORM labelling, possibly use a more optimal STORM buffer see eg Olivier et al PLoS One 2013, and acquire and analyse more optimal test data. An online resource with several experimental datasets from well labelled samples is also available:<http://bigwww.epfl.ch/smlm/datasets/index.html>. Possibly you could also ask Susan Cox for her raw data used in the original 3B paper.

Reply: Please accept our sincere thanks for referring to the EPFL dataset. We have included the results on the EPFL dataset in the main paper. We have also included the results on test data provided by Dr. Susan Cox on the 3B project page.

Please see the newly added **Figs. 3 and 4** (Pages 6 and 8, respectively) and **magenta colored text on Pages 2, 4, 5, and 7** in the revised main paper.

Comment 4: The run time of MUSICAL vs 3B is not clearly compared. Since the main advantage claimed on pg 6 is that MUSICAL is basically 3B performance but faster, this is really important. In Fig S19, there is a table of run times with 3B as 72 s vs MUSICAL 94 s. I think 3B was run on different hardware (cloud computing) which is why it is quite fast, but this is not clearly discussed. It is important to see a proper comparison of run times on the same hardware.

Reply: We have included the computation time of 3B when executed on the same system as MUSICAL in Fig. 4(e) of the main paper. Thank you for the suggestion.

Please see **Fig. 4(e)** on Page 8 of the revised main paper and **blue colored text on Page 7**.

Comment 5: There is no discussion of performance of the algorithms as a function of emitter density (bright emitters/ μm^2) which is the key parameter in HD algorithm performance.

Reply: In the opinion of the authors, emitter density provides useful insight into localization algorithms which use one frame at a time. However, in the context of the multi-frame methods, emitter density alone cannot represent the performance of these methods and has to be studied essentially in conjunction with the blinking rate and the net spatio-temporal sparsity. The work requires a separate analytical framework and an extensive treatment which deviates from the main proposition of the paper. However, we assure that such analysis will appear in subsequent publications and will be given an elaborate treatment.

Comment 6: The Shaevitz lab published an algorithm SCORE (Deng et al, PLoS One 2014), which on the surface seems reasonably similar algorithmically - principle component/ eigenvector analysis of multiple frames to extract super-resolved image. I am curious about how the two algorithms differ. SCORE achieves only quite low resolution (~ 100 nm) rather than the high resolution claimed here.

Reply: We thank the reviewer for bringing up the question of differences between SCORE and MUSICAL. We have discussed the differences between MUSICAL and SCORE in **section VI.B on Page 15 of the revised supplement**. Below, we explain the differences:

- Score uses only those eigenimages that belong to the range of the measurement matrix. On the other hand, the use of eigenimage in the null space is the core of MUSICAL and is even more important than the use of eigenimages in the range.
- MUSICAL uses the information of PSF not only for computing the projections on the range and the null space but also to incorporate sliding window function, which is critical in reducing the impact of noise and other emitters distant from the emitters in the sliding window. SCORE does not use any sliding window; its use of PSF is limited to the computation of the projections onto the eigenimages in the range.
- SCORE casts the reconstruction as a minimization problem of minimizing the difference between the measured and estimated covariances after estimating an initial covariance matrix using an exponential indicator function of the projection. MUSICAL does not need to solve an iterative optimization and is expected to be more computation efficient for this reason.

Comment 7: Line plots are not an ideal metric for resolution (as discussed eg in supplement of Legant et al Nat Meth 2016). I realize FRC is not necessarily straightforwardly applicable for non-localization based methods. Do the authors have any other alternative measures of resolution? What about a radial plot of the reconstructed image fourier transform - essentially the system MTF?

Reply: We thank the reviewer for the suggestion of considering an alternative measure. We found that a suitable representation of resolution for multi-frame methods such as MUSICAL, similar to MTF, could be through the plot of contrast as a function of the distance between two lines of emitter when the two lines of emitters have the same emission statistics and the statistics do not vary with the distance between the emitters. This plot is now included in Fig. 1(e) of the main text.

See Fig. 1(e) on Page 3 and blue colored text on Page 4 of the revised main paper. Please see the section III.D in blue colored text on Page 9 of the revised supplement.

Comment 8: The claim pg 6 "Lastly, we note that to the best of our knowledge, none of the currently known super-resolution algorithms so far has demonstrated the capability to produce super-resolved images of live cells in their natural environment with total acquisition times of the order of 50-250 milliseconds using regular nonspare blinking dyes" is wrong - see Huang et al Nat Meth 2013.

Reply: Thanks to your comment, we realize that we should highlight that we have not used photo-switchable dyes or oxygen scavenging system such as used in Huang et. al. We have included this information in the revised main text, see blue colored text on Page 8 of the revised manuscript.

Please see blue colored text on Page 9 of the revised main paper.

We hope that the reviewer finds the revised manuscript satisfactory.

With thanks and best regards,

Authors

Replies to the comments of Reviewer #2:

Dear reviewer,

We thank you for finding the manuscript well-written and interesting. We also thank you for several interesting and constructive queries, which have helped explain our method better. Below, we reply to your questions and detail the changes made in the manuscript to incorporate your suggestions.

Comment 1a: The method resembles another version of noise reduction + deconvolution.

Comment 1b: I did not see how this method would provide single molecule location information which is the key for high resolution PALM/STORM images.

Comment 1c: I am not sure how the blinking behavior of the single molecules helped

Reply: While MUSICAL may be inferred to be a form of deconvolution, it should be noted that MUSICAL is unlike single molecule localization techniques which localize emitters in each frame and then collate the localizations in all the frames. MUSICAL instead relies on the temporal variations in blinking behavior of molecules, which results in temporal fluctuations in the measured intensity at a pixel. These fluctuations over time in image plane are represented through eigenimages. Thus, all the frames participate together to form the eigenimages and hence MUSICAL cannot be applied on a single frame at a time.

While the eigenimages do represent the patterns in the image stack, they themselves are incapable of providing super-resolution. Further, MUSICAL does not perform deconvolution in the conventional sense since it uses splitting of the eigenimages into range and null spaces and relies more on the projection upon the null space rather than the projection on the range.

Thanks to your query, these aspects are explained more explicitly in Fig. S6 and section III.A of the revised supplement. *Please see Fig. S6 on Page 8 and the blue colored text on Page 7 of the revised supplement.*

Comment 2a: I am not sure how the blinking behavior of the single molecules helped here except that it allows the authors to apply a soft window on image sequences to improve the computation speed.

Comment 2b: For example, you can get a sharper profiles by taking a square of a confocal image. The "squared" images look sharper but it will not resolve features that are previously unresolvable.

Reply: Blinking has no consequence on the soft window at all. Soft window is applied in the spatial domain and thus blinking has no influence on what data is retained in the soft window. However, the point spread function and the locations of the molecules in the window do influence what data is retained in the window.

We highlight that blinking is a basic underlying requirement for super-resolution in all multi-frame methods, including MUSICAL. Other multi-frame methods include 3B, SOFI, and ESI. Blinking is also essential for single molecule localization microscopy techniques such as STORM. Not only should the particles blink, but the blinking should be independent. It is the blinking of the particles that introduces differentiability of two particles which are otherwise unresolvable. We show how blinking is used in MUSICAL and what would happen if blinking is absent through the aforementioned example of eigenimages.

Thank you for highlighting that the role of blinking might not have been clear in the previous version. Please see blue colored text on Page 7 and Fig. S6 on Page 8 of the revised supplement.

Comment 3a: Another point is, taking the ratio between the projections in the two subspaces will artificially shrink the FWHM of structures and therefore, line profiles should not be used to demonstrate resolution achieved.

Comment 3b: As mentioned above, line profiles should not be used to quantify resolution in highly processed images.

Reply: This is a point raised by two other reviewers as well and we thank you all for raising this concern. Since modulation transfer function or Fourier ring correlation are not directly applicable for multi-frame methods, we devised an alternative metric similar to modulation transfer function. We plot of contrast as a function of the distance between two lines of emitter when the two lines of emitters have the same emission statistics and the statistics do not vary with the distance between the emitters. This plot is now included in Fig. 1(e) of the main text.

See Fig. 1(e) on Page 3 and blue colored text on Page 4 of the revised main paper. Please see the section III.D in blue colored text on Page 9 of the revised supplement also.

Comment 4: For example, you can get sharper profiles by taking a square of a confocal image. The "squared" images look sharper but it will not resolve features that are previously unresolvable. Therefore to demonstrate the resolution achieved, the authors should provide super-resolution images resolving features that are previously obscured by diffraction, for example, two sides of microtubules, mitochondria cristae, or vesicles.

Reply: In addition to Fig 1(e) which directly addresses resolvability of two lines, we show in Fig. 3(a) of the main paper that MUSICAL can resolve optically unresolved microtubules. Thank you for the suggestion.

Please see Fig 3(a) on Page 3 and magenta colored text on Page 4 of the revised main paper.

Comment 5a: The MUSICAL images from Figure 2a-2, there's a dark region around bright filaments. Is this real or is image artifact.

Comment 5b: Figure S18c also showed image artifact that were not part of the original signal. Would there be a way to decrease the large amount of artifact introduced?

Reply: The effect mentioned by you is the consequence of side lobes of MUSICAL, which were discussed in Figs. S6 and S13, and section V.D of the previous version of the supplement (Fig. S9 and S16, and section V.D of the revised supplement).

We have included a process for reducing such artifacts in section V.I on Page 15 of the revised supplement. Please also see the revised Fig S21 (originally Fig. S18 of the supplement) and the newly added Fig. S22 on Pages 14 and 15, respectively, in the revised supplement.

Thank you for the comment 5b, since it helps in adding another interesting aspect to the paper.

Comment 6: Images from Figure 2 b2,c2,d2 of actin filaments are quite different from what others have shown using PALM/STORM (Nat Methods. 2012 Feb; 9(2): 185-188.). Is there a reason why it would look different from images that others have reported?

Reply: The images of actin filaments in 'Nat Methods. 2012 Feb; 9(2): 185-188' are from a cell sample, comprising of cortical cytoskeleton and actin bundles. On the other hand Figure 2 in the main paper corresponds to in vitro preformed actin filaments. The structures are quite different in these different scenarios. Further, PALM and STORM perform single molecule localizations whereas MUSICAL does not perform localizations. It reconstructs the details of the structures captured through the blinking of fluorophores on the structures. In this sense, it is similar to fluctuations based multi-frame microscopy techniques such as 3B, ESI, and SOFI, although aiming towards resolution closer to STORM/PALM. We have included comparison of MUSICAL with these fluctuation based methods as well as STORM and PALM using experimental data used for benchmarking STORM and PALM methods in Figs. 3 and 4 of the revised manuscript.

Please see Fig. 3 on Page 6 and Fig. 4 on Page 8 of the revised manuscript.

Comment 7: Equation 3 in the supplement is derived based on the assumption that the pixel size of the imaging system is much smaller than the PSF width. This might not be true as many PALM/STORM systems including commercial systems use a pixel size that are close to the half width of the PSF.

Reply: This assumption is not very restrictive, as seen in the experimental results in which the entire PSF is represented by 7 pixels and the FWHM of about 175 nm (for emission wavelength 510 nm) is approximately 2.7 times the pixel size of 65 microns.

Comment 8: The authors should double check the usage of the phrase "in vivo" in the context of the manuscript. Would "live cell" be more accurate?

Reply: Thank you for the suggestion. We have incorporated it.

We hope that the reviewer finds the revised manuscript satisfactory.

With thanks and best regards,

Authors

Replies to the comments of Reviewer #3:

Dear reviewer,

Thank you very much for your generous appreciation of our manuscript and giving insightful comments. Here, we reply to your comments and detail the changes made in the manuscript to incorporate your suggestions.

Comment 1.1: It seems the authors pick the indicator function by intuition, is there other better way to do it?

Reply: Yes, the choice of the indicator function is heuristic but based on the following requirements:

- The indicator function should make use of the null space and should satisfy eq. (20) in the supplement.
- It should not inhibit in stitching the images of the sliding windows.
- It should be robust to the presence of noise.

There may be more indicator functions that satisfy the above requirements and have other salient properties. This investigation is relegated to future research on MUSICAL.

Comment 1.2: In the description of α , the authors say that the larger the α the narrower the spread of the fluorescent molecule, which implies better resolution. However, the authors only choose $\alpha=4$ and stop increasing the number. Why not keep increasing to get even better resolution?

Reply: We thank you for raising concern over the role of the parameter α . We have included a detailed discussion on the role of α in the newly added section II.F of the revised supplement on Page 6.

Comment 1.3a: It seems to correctly estimate a threshold σ for signal noise separation is important to have a nice reconstruction. In the main paper, the authors estimate the SNR of sample 1,2 and in-vivo experiment. How do authors estimate SNR from the data?

Comment 1.3b: In addition, if we do not know the SNR, the authors say that they recognize the knee structure of the singular-value plot to set the threshold σ . However, Fig. S4 shows the reconstruction will have better resolution when you set the threshold even below the knee structure. I am wondering if you have a clear way to set the threshold for the case without SNR? Or the authors always estimate the SNR and set the threshold?

Reply: Thanks to your comment, we realize that our mention of the SNR for computation of the thresholds in the experimental images may be misleading or confusing. We have thus clarified it in the red colored text on Page 16 of the revised supplement.

Further, about the threshold used in Fig. S4, the knee appears at σ_4 . As long as the threshold σ_0 is between σ_4 and σ_5 , the range defines by σ_0 contains 4 eigenimages, which corresponds to 4 patterns from 4 independently emitting emitters. We realize that our statement ' σ_0 is chosen as the the approximate value where a knee feature is observed in the logarithmic plot of singular values' must be the source of confusion. We have corrected this sentence as: ' σ_0 is chosen to be slightly less than the value

where a knee feature is observed in the logarithmic plot of singular values'. Please see the **red colored text on Page 6** of the revised supplement.

Comment 1.4a: In the main paper, the authors use a sliding window of the size of 7x7, which corresponds to the size of PSF, so the maximal rank of each patch is 49. This is why the authors only collect 49 images for reconstruction. I am curious about how the number of pixels for each window will affect the reconstruction. If the authors use a window size of 10x10, for example, and take 100 images. Will the reconstruction be better than the previous case?

Comment 1.4b: It would also be interesting to see what would be the effect of number of frames to reconstruction in the simulation.

Comment 2.1a: Following question 1.4, the last two columns of Fig. 2 in the main paper compared the reconstruction using 10000 frames and 49 frames. Although the reconstruction with 49 frames capture the main structure of the sample, the fine features seem to look different from the one with 10000 frames. Maybe some of the fine-structure information is not contained within these 49 images?

Comment 2.1b: It may also suggest MUSICAL does not have uniform resolution performance across the field of view?

Comment 2.2: For Fig. S7, it may seem that the reconstruction with 49 frames is sharper than the one with 200, 400, or 800 images. However, the one with 49 frames look not well-sampled (some parts of the sample is missing). Maybe 49 images are too few to have a nice reconstruction?

Reply: We have included the effect of window size in the *newly added section II.D on Page 5* of the revised supplement.

You are right about the potential absence of information in the selected subset of frames and consequently the information being missing in the reconstruction. We have explicitly discussed this in **red colored text on Page 4** of the main paper.

The field of view has no consequence on MUSICAL, assuming uniform distribution of the excitation power in the field of view.

Comment 1.5: MUSICAL requires PSF to be known. Do the authors calibrate the PSF? What is the tolerance of the algorithm if the known PSF does not well match the experiment PSF?

Reply: As mentioned in the original version of the paper and the supplement, we used computed Airy function as the PSF. We mention it again in **section VI.B.1 of the supplement**, which discusses the implementation details of MUSICAL.

We thank the reviewer for raising the concern of mismatched PSF. *We have addressed this using a newly added section II.G and Fig. S4 on Page 6* of the revised manuscript.

Comment 3.1: Supplementary section III C, the abbreviation SBR shows up before the definition in section III E.

Reply: Thanks for pointing this out. We have addressed this.

Comment 3.2: Supplementary III D, is there any reason that authors choose different and for different simulations?

Reply: For SynEx1, SynEx2, SynEx3, SynEx4, and SynPeriod, the choice is heuristic. For SynFork1 and SynFork2, the choice is to allow the data to be suitable for STORM too.

Comment 3.3: Supplementary IV A, what does "0.4 times of the 99.9% of the histogram of MUSICAL intensities" mean? There are several descriptions like this.

Reply: We understand and apologize for the confusion. Since similar criterion is used throughout the paper, we have explained it in detail in section VI.C.7 in blue colored text on Page 16 of the revised supplement.

We hope that the revised manuscript is satisfactory for the reviewer.

With thanks and best regards,

Authors

Replies to the comments of Reviewer #4:

Dear Reviewer,

Thank you for appreciating the manuscript. Addressing your comments help in improving the readability of the paper and explaining the concepts better. Here, we reply to your comments and detail the changes made in the paper.

Comment 1: I believe that the basis of the method needs to be much more clearly explained if publication in this journal is to be considered. I particularly struggle with understanding what the range and null spaces are and what the eigenimages might look like. This could potentially be assisted with an additional supplemental figure showing examples of eigenimages which belong to each space.

Reply: Thank you very much for the suggestion. *We have included eigenimages for the example SynEx1 in the newly added Fig. S6 on Page 8 and discussed them in blue colored text on Page 7 of the revised supplement. Please also see the newly added section VII on Page 17 of the revised supplement for a discussion on the concepts of the range and the null space.*

Comment 2: I do not believe that the authors have clearly established WHY the performed manipulations are expected to give super-resolution. My very simplistic interpretation is that the eigenvector decomposition will, somewhat like independent component analysis (Lidke et al 2005), give a series of eigen-images which represent the individual molecules present, and that the projection onto the shifted copies of the PSF is somewhat like using an autocorrelation to find molecule peaks.

Reply: We hope that the projections of the PSF on the eigenimages in Fig S6 make these aspects clear.

While eigenvector decomposition is analogous to independent component analysis, the use of eigenimages corresponding to zero singular values (equivalent to the independent components orthogonal to the data) in MUSICAL is different from most other linear algebraic treatments of the data. Correlations (including auto correlations) essentially address only the range of the matrix (i.e. eigenimages with non-zero singular values). The use of null space is what distinguishes MUSICAL from other techniques and helps in providing better resolution than correlation based techniques.

Comment 3: The test function, however, seems fairly empirical, and the alpha parameter is not well justified. I find the alpha exponent very suspicious, as it seems to be chosen purely to make the images look sharper. You can make any fluorescence image look sharper by raising it to a power, but this would never be considered super-resolution.

Reply: These concerns have been raised by other reviewers as well.

Yes, the choice of the indicator function is heuristic but based on the following requirements:

- The indicator function should make use of the null space and should satisfy eq. (20) in the supplement.
- It should not inhibit in stitching the images of the sliding windows.
- It should be robust to the presence of noise.

We thank you for raising concern over the role of the parameter α . We have included a detailed discussion on the role of α in the newly added section II.F of the revised supplement on Page 6.

Comment 4a: One of the major issues with other non-localization based approaches (specifically SOFI) is that they are non-linear in emitter concentration. Is this method linear (some of the simulated data suggests that it might not be with the crossings of lines appearing more than twice as bright as a single line)?

Comment 4b: The resulting reconstructions of real data seem even spottier than PALM/STORM. Might this be a result of non-linearity, or is there another issue with the method?

Reply: You are correct about the non-linearity of MUSICAL. MUSICAL is non-linear due to the presence of d_{PN} in the denominator of the indicator function and the use of the parameter α . We hope that the discussion in section II.F of the revised supplement addresses this aspect well.

Comment 5: The width of a line profile (as used to quantify resolution on experimental data) is useless if the method is non-linear, and is generally somewhat suspect even for linear methods. Much better is to show that two lines can be resolved. The data in Fig1 on this does not show the two lines being clearly resolved at better than the diffraction limit.

Reply: This is a point raised by two other reviewers as well and we thank you all for raising this concern. Since modulation transfer function or Fourier ring correlation are not directly applicable for multi-frame methods, we devised an alternative metric similar to modulation transfer function. We plot of contrast as a function of the distance between two lines of emitter when the two lines of emitters have the same emission statistics and the statistics do not vary with the distance between the emitters. This plot is now included in Fig. 1(e) of the main text.

See Fig. 1(e) on Page 3 and blue colored text on Page 4 of the revised main paper. Please see the section III.D in blue colored text on Page 9 of the revised supplement also.

Comment 6: A more extensive comparison to other methods this approach could address an important niche in the analysis of blinking based super-resolution - namely the analysis of dense data in a way that is a) better than SOFI and b) faster and less reliant on metaparameters than 3B.

Reply: We thank you for the suggestion of comparison with other methods. Accordingly, we have included comparison of MUSICAL with SOFI, 3B, and ESI using independent experimental data from the EPFL's SMLM datasets and 3B test data. We thank you for this valuable change in the manuscript.

Please see the newly added Fig. 4 on Page 8 and magenta colored text on Page 7 of the revised paper.

We hope that reviewer finds the revised manuscript satisfactory.

With thanks and best regards,

Authors.

Reviewers' Comments:

Reviewer #1 (Remarks to the Author):

The additional analysis on high quality experimental data shows comparable resolution to 3B, and significantly better performance compared to the other methods, and the run time analysis is much better compared with 3B. The data now supports the claim the algorithm is a significant improvement on state of the art, and I recommend publication.

While the paper was in review, Henriques SSRF method (another multi-frame high density blinking algorithm) was published in Nat Comms - I do not see this as any sort of issue but they will require comparison in future work.

Minor issues:

- Scale bars in Figs 2-4 are incredibly obtrusively placed - they should be smaller in many cases and always in a corner, they make it difficult to read the figures.
- Fig 4c the ordering of the sub figures is pretty random and makes it hard to find the different softwares and compare between them.
- Fig 5e - the "distance from closest maximum" plot is very confusing and probably less reliable than a single line profile, it should probably be replaced by one or more line profiles

Reviewer #2 (Remarks to the Author):

In the revised manuscript, Agarwal et al. added significant new data demonstrating the resolving power of the developed methods as well as additional explanations for the algorithm. I can now see the potential of this method but remain concerned about the artifacts (side lobes) that it may generate. Suppression of these side lobes will be essential for correctly interpreting the resulting images and quantification of the resulting data. In this respect, the authors used a non-zero threshold to mask these low intensity lobes. However, this might only work on isolated structures (i.e. line or dot like structures) where side lobes will not overlap and enhance each other. I therefore remain concerned about the usefulness of this algorithm given artifacts showing throughout the manuscript (side lobes and as well as discontinuous line structures even when continuous structures, Fig S11, Fig S10, Fig S8, Fig 1, C3 (C1)). However, I remain positive on and quite impressed by the potential of this algorithm but urge the authors to eliminate these artifacts before publishing their development.

Reviewer #3 (Remarks to the Author):

[The reviewer believes the paper is now ready for publication]

Reviewer #4 (Remarks to the Author):

The authors have made a reasonable attempt to address the reviewers critiques, which is not an easy task given 4 different reviewers with different expectations.

Despite these efforts, I still have some reservations / skepticism about the technique. These principally relate to the effect of non-linearity (3B and deconvolution / compressed sensing based methods are approximately, if not perfectly linear in dye concentration, and the negative aspects of non-linearity cannot be understated), and the *strict* validity of the resolution measurements. That

said, measuring resolution is a surprisingly difficult task and similar criticisms could be leveled at a lot of papers in the field. I now have some idea of how the projection on range and null spaces, and their comparison might work to give rise to super-resolution, but still don't understand it completely. Because of these remaining concerns, I wouldn't be rushing to try and implement the technique in my lab (i.e. I'm not yet fully sold on it). I do feel that there is enough in the paper to warrant publication, it is not misleading, and that people within the field will benefit from the information contained. I don't think that there is much to be gained by requesting additional comparisons or experiments.

My remaining concern is clarity. The paper started out being fairly long and complicated, and has not got clearer or easier to understand on addition of the changes in response to the reviewers. There is a *huge* amount of information in the paper and it can be hard to follow at times, leading to information overload. Ultimately I think this will hurt citations, and I would strongly recommend simplifying where possible and relegating much of the material to supplementary information. My preference would be for each figure to have about half the number of panels, and for the figures and text in the main manuscript to focus on what the key messages are. In my opinion WHY the method works deserves more emphasis than the actual performance or comparison to other methods.

The supplement is also dense, and as there are (to my knowledge) no formatting constraints on supplementary information, I'd recommend using a slightly less dense layout here with a clearer separation into sections.

Reply to reviewers

Dear Reviewers,

We thank you for the valuable time and effort invested on the manuscript. We are extremely pleased with the constructive and favorable review. We address your outstanding concerns below:

Reviewer 1: SRRF method

Authors: We will be glad to compare MUSICAL with SRRF in future publications.

Reviewer 1: Minor formatting comments

Authors: We have tried to address them with the help of editorial suggestions.

Reviewer 2: Side lobes.

Authors: We did include a more sophisticated scheme for side lobes suppression (now Supplementary Note 9) in the previous version. However, we missed cross-referencing this scheme appropriately, such that it can attract reader's attention. We have corrected this problem (see Supplementary Figure 14).

Reviewer 4: Non-linearity

Authors: The work to adapt MUSICAL into a reconstruction method, such that the super-resolved image is not a map of the MUSICAL indicator function but a map of emitter locations and emissions, is on-going. We hope that the adaptation can be published in the future.

Reviewer 4: Clarity and density of information

Authors: We have reduced the amount of information in the figures in the main paper, moving redundant information to the supplement as much as possible. Arranging the supplement in the form of almost Supplementary Figures and associated Supplementary notes has helped in making the supplement less dense and clearer. We thank your suggestion and the editorial feedback on the manuscript which has helped us improve the readability.

We thank the reviewers once again for their precious time.

Best regards,

Authors